# Adjuvant Chemotherapy in Elderly Colorectal Cancer Patients

**DOI:** 10.3390/cancers12082289

**Published:** 2020-08-14

**Authors:** Bengt Glimelius, Erik Osterman

**Affiliations:** 1Department of Immunology, Genetics and Pathology, Uppsala University, SE-75185 Uppsala, Sweden; erik.osterman@igp.uu.se; 2Department of Surgery, Gävle Hospital, Region Gävleborg, SE-80187 Gävle, Sweden

**Keywords:** colorectal cancer, adjuvant chemotherapy, elderly

## Abstract

The value of adjuvant chemotherapy in elderly patients has been the subject of many overviews, with opinions varying from “not effective”, since randomized trials have not been performed, to “as effective as in young individuals”, based upon many retrospective analyses of randomized trials that have included patients of all ages. In the absence of randomized trials performed specifically with elderly patients, retrospective analyses demonstrate that the influence on the time to tumour recurrence (TTR) may be the same as in young individuals, but that endpoints that include death for any reason, such as recurrence-free survival (RFS), disease-free survival (DFS), and overall survival (OS), are poorer in the elderly. This is particularly true if oxaliplatin has been part of the treatment. The need for adjuvant chemotherapy after colorectal cancer surgery in elderly patients is basically the same as that in younger patients. The reduction in recurrence risks may be similar, provided the chosen treatment is tolerated but survival gains are less. Adding oxaliplatin to a fluoropyrimidine is probably not beneficial in individuals above a biological age of approximately 70 years. If an oxaliplatin combination is administered to elderly patients, three months of therapy is in all probability the most realistic goal.

## 1. Introduction

Globally, colorectal cancer (CRC) is the third most common cancer and the fourth most common cause of cancer death [1] {Arnold, 2017 #15794}. The median age of patients in most Western countries is around 70 years. Thus, approximately half of the patients belong to an elderly group, 70 years of age or older.

At diagnosis, slightly more than 20% of the patients have distant metastases and a further 20–25% will recur after curative surgery. The latter group is the target of adjuvant chemotherapy, where the purpose is to decrease the risk of recurrence by eradicating subclinical tumour deposits which remain in the body after surgery. The recurrence risk determines the need for adjuvant therapy. One crucial question in elderly patients is whether the recurrence risk is the same as that in younger individuals. Population data seldom provide reliable reports of recurrence risks, and overall survival (OS) or disease-free survival (DFS) is often presented indicating the need. Since mortality is higher in the elderly, it may falsely appear as if the need for adjuvant treatment is higher. A second crucial question is whether the adjuvant therapy that works in younger adults is as effective in the elderly. Without randomized trials specifically encompassing the elderly, our knowledge is based upon retrospective analyses of elderly patients included in trials.

This review will summarize what the retrospective analyses have shown. In addition, it will discuss an aspect not usually properly considered, namely, the need of adjuvant chemotherapy in elderly patients compared with that in younger patients.

### 1.1. Which Patients Are Elderly?

There is greater heterogeneity in physical, mental, and cognitive health in elderly patients than in younger patients. Thus, in the elderly, biological age is more relevant than chronological age [2,3,4].

There is no universal definition of who is elderly and who is not. The World Health Organization (WHO) definition varies substantially in different areas of the world and it is not easy to find a proper definition, but individuals over an age of 65 are frequently considered elderly [5]. Most adjuvant CRC studies use 70 years [2,3,4] and some have used 75 years as a cut-off [6,7]. When reviewing the literature, one has to stick to available data in the publications. In some trials, an upper age limit has sometimes been defined (70–75 years), limiting the number of elderly patients. However, even if no upper age limit was present, the number of individuals belonging to an elderly group is limited due to extensive selection. In all probability, only a small fraction of the fittest elderly is included.

### 1.2. How to Measure Benefit? The Scientific Evidence

Adjuvant treatment improves DFS and OS by reducing recurrences. The effect on recurrences is best recorded as a reduced number of recurrences or an improved time to recurrence (TTR). However, in trials, DFS (including recurrences, any death and second malignancies as events) [8] and OS are recommended [9] since they also account for negative side-effects, which TTR ignores. Adjuvant treatment is, however, generally only clinically motivated if it improves DFS or possibly even OS, balancing gains and losses. Because of the comparatively limited gains from adjuvant treatment in CRC, although clinically meaningful in many patients, only randomized trials can confirm whether a treatment is efficacious and motivated.

There are no randomized trials performed specifically in elderly patients to answer the second most crucial question raised above, i.e., whether adjuvant therapy prevents recurrences sufficiently well in elderly patients to motivate the increased risks of toxicity seen in these patients. This has led to one conclusion that adjuvant treatment in elderly patients is not evidence-based [10].

There is thus limited formal scientific evidence of sufficient gains in elderly patients, apart from that culled from retrospective analyses of the fittest elderly patients included in trials. The evidence has been summarized or at least discussed multiple times [11,12,13,14,15,16,17,18,19,20,21,22,23,24]. Much of the retrospective evidence has been created by the Adjuvant Colon Cancer Endpoints (ACCENT) collaborative group [25]. This group has collected individual patient data from a large number of clinical randomized trials run during the past 40 years and has thus been able to shed light on research questions requiring pooling of data in less frequent subgroups, such as elderly patients. It should be noted that the ACCENT database does not contain data from all clinical trials, as the Cochrane Library [26] aims at achieving.

One study from the ACCENT group reported that elderly patients included in the trials surviving five years without recurrence had a better survival than individuals from a matched general population, emphasizing that patients included in the trials are fit individuals [27].

## 2. What Is the Present Need in Elderly Patients?

This question falls into two parts, firstly, whether the need in elderly patients is less today than it was in the past and, secondly, whether the need differs in younger and older adults. At least as regards colon cancer, recent evidence suggests that recurrence risks, clinically dominated by distant metastases, have decreased compared with the trials with a surgery-only group, proving that adjuvant chemotherapy works. We will first describe evidence that recurrence risks after radical surgery are less today than in the past [28,29,30,31] and then show present recurrence risks according to age in the Swedish population of colon cancer patients having revealed apparently lower recurrence risks than in the past [32,33]. To the best of our knowledge, we have not seen any similar reports on these topics.

### 2.1. Evidence of a Lower Need for Adjuvant Therapy in Elderly Patients Today Compared with the Situation in the Past

#### 2.1.1. Colon Cancer

Fewer recurrences were reported in two investigations analysing outcomes in the ACCENT database which summarized the evidence from trials that included patients between 1978 and 2007. One study analysing recurrence risks in elderly patients stated that “many calculators (of DFS/OS) reflect older practices” [30] and “called into question historical data related to the benefit of FU-based adjuvant therapy in such (stage II) patients” [28]. When analysed according to age, a significant interaction (*p* = 0.009) was seen with stage migration only in patients older than 75 years. The present guidelines [34,35,36,37], based on the results of the older trials, are thus no longer relevant, at least not in early stages where they overestimate the present need for adjuvant treatment, especially in elderly patients.

Furthermore, in a recent update from the ACCENT database [38], stage III patients treated with an oxaliplatin/5-fluorouracil (5-FU) regimen (FOLFOX or FLOX) and included during a more recent period (2004–2009) compared with older studies (1998–2003, the MOSAIC trial [39] and NSABP C-07 using bolus FLOX [40]) fared better. TTR was significantly better in the new era, hazard ratio; 95% confidence interval (HR; 95% CI 0.83; 0.74–0.92). This improvement was particularly pronounced in patients above 65 years of age (HR; 95% CI 0.71; 0.58–0.86) and not seen in those under 45 years of age. DFS behaved similarly. OS was also improved but this was mainly ascribed to longer survival after recurrence, median 2.2 years vs. 1.5 years, *p* = 0.000. This apparently improved outcome in elderly patients is in all probability not caused by fewer recurrences, as in the two previous analyses of the ACCENT database [28,30], but rather reflects improved possibilities of treating elderly patients with an oxaliplatin combination.

#### 2.1.2. Rectal Cancer

The risk of local recurrence has in rectal cancer decreased thanks to the use of pre-operative (chemo)radiotherapy (RT/CRT), precise surgery, improved preoperative staging using magnetic resonance imaging (MRI), and multidisciplinary team meetings [41]. Systemic recurrences have only marginally decreased [42] and now contribute to the majority of rectal cancer deaths after radical surgery. The risk of (systemic) recurrence is about 20% in multicentre series [43] and in nationwide populations [42] when patients are adequately staged and treated, providing the potential gain from adjuvant treatment. To refer to an OS of 50–60% after five years to indicate the need is inappropriate. Elderly patients have not been analysed separately, and there are no indications that recurrence patterns differ from those observed in younger rectal cancer patients.

In rectal cancer, the issue is not only if adjuvant treatment prevents recurrences sufficiently well in the elderly but also whether it is effective in younger adult patients. The discrepancy in the efficacy of adjuvant chemotherapy observed in colon and rectal cancers has been the topic of many articles [44,45,46,47].

### 2.2. Are Recurrence Risks Today Different in Elderly and Young Patients?

Although most colorectal cancers are adenocarcinomas showing different degrees of differentiation and variable amounts of mucinous expression, great heterogeneity is molecularly seen between different cancers at different locations of the large intestine [48,49]. The relative distribution of some of these changes differs according to age. Of immediate interest for evaluating the risk of recurrence and potentially also for the response to fluoropyrimidines are deficiencies in the expression of mismatch repair proteins (dMMR), often resulting in microsatellite instability (MSI). Most tumours with dMMR are right-sided, and right-sided tumours are more common in the elderly, especially in elderly women. In a large series of Japanese patients, the proportion of right-sided tumours increased with age from less than 30% in those under 65 years of age to almost 50% above 85 years of age [50,51]. MSI tumours are associated with a smaller risk of recurrence and, at least in some, but not all studies [52,53], a poorer response to fluoropyrimidines. Since MSI tumours are rather infrequent, about 15% of newly diagnosed cases but as high as 35% in patients above 85 years [54,55], this may decrease the need in elderly and lessen the effect of adjuvant chemotherapy. Very old patients generally have less advanced tumours with less node-positivity than younger patients [56], indicating a more limited need for adjuvant treatment. *BRAF*-V600E-mutated tumours are also more common in the elderly but display an increased risk of recurrence; overlapping is seen to a great extent between MSI-high (MSI-H), *BRAF*-V600E mutations, right-sidedness, and high age [56,57]. However, in a Dutch study [58], MSI and *BRAF* mutations were not more common in elderly than in young individuals. The poor prognosis of *BRAF*-V600E mutations in stage II + III colon cancer in most studies seems to be overridden by the good prognosis seen with MSI-H [59,60].

In several clinical trials and observational cohorts, MSI-H and right-sided tumours were independent predictors of improved OS in stage II + III colon cancer, whereas *BRAF*-V600E mutations were not [61]. However, the effects of including these molecular changes were modest when compared with routine clinico-pathological variables. Thus, even if molecular alterations in primary CRC demonstrate differences when the tumours of young and old patients are compared, these are not of such a magnitude that they put into question the general statement that recurrence risks do not differ between age groups.

In a study by Hoshino et al. [62], exploring risk factors for recurrence in 1354 patients above 70 years of age, the same factors were important in both elderly and younger individuals.

### 2.3. Recurrence Risks According to Age in Swedish Colon Cancer Patients

In order to illustrate the present needs, an analysis of recurrence risks in the Swedish population for colon cancer is presented for different age groups in relation to whether adjuvant chemotherapy was administered or not in stages II and III.

Sweden has a public health care system with equal access for all citizens. In CRC, several clinical trials have been run, many quality assurance measures have been taken during the past 20 years, and a nationwide quality registry, SCRCR, allows excellent control of whether the measures result in expected good results uniformly distributed across the country. We recently published recurrence risks that are apparently lower than they were in the past, i.e., when the trials showed that adjuvant chemotherapy lowers recurrence risks [32]. As can be seen in Table 1, recurrence risks in patients not treated with adjuvant therapy, who make up the majority of stage II patients irrespective of age, are around 10% for all ages above 50 years.

The same lack of difference according to age was seen also in stage III, where almost all patients up to 70 years and the majority between 70–79 years of age received adjuvant therapy. To investigate whether differences were statistically significant, a Cox proportional hazards regression was calculated adjusting for stage and adjuvant treatment. Patients aged between 70–79 and 80–89 years old fared worse than the reference group of patients < 50 years old (HR 1.3, *p* = 0.005 and 0.006). No statistically significant differences were seen for patients aged between 50–59, 60–69 or those over 90 years of age. The higher recurrence risks in patients with stage II receiving adjuvant treatment reflect the selection according to the presence of risk factor(s).

## 3. What Do the Trials Tell Us about the Efficacy of Adjuvant Therapy in Elderly Patients?

### 3.1. Colon Cancer

#### 3.1.1. Treatment with Fluoropyrimidines Alone

In two pooled analyses of seven colon cancer trials [12,13] exploring the value of adjuvant 5FU-based treatment versus surgery alone, the same benefit was seen in patients above or below 60 [13] or 70 years [12]. Toxicity in 15% of elderly patients above age 70 also appeared to be similar (Table 2).

In the QUASAR study [63], which mainly included patients with low-risk stage II disease, no benefit of adjuvant chemotherapy among patients > 70 years old was observed. In a Dutch study [64], where the gains from adjuvant therapy were confirmed, no differences in efficacy (RFS or OS) were seen between patients below age 60, between 60–65, and between 65–75 years, although RFS tended to be poorer in those individuals over 60 years of age.

In the non-inferiority trial X-ACT [65] which compared capecitabine with bolus 5FU/leucovorin (Lv) according to the Mayo clinic schedule, non-inferiority was shown irrespective of age with a cut-off at 70 years. In two Japanese non-inferiority trials [66,67] where UFT/Lv were compared with 5FU/Lv and S1 (combination of tegafur, gimeracil, and oteracil) with UFT/Lv, the results were statistically similar according to age, although S1 appeared to result in better DFS than UFT/Lv in those above 70 years of age.

#### 3.1.2. Treatment with a Fluoropyrimidine and Oxaliplatin

In the MOSAIC trial [39], the addition of oxaliplatin (FOLFOX-4) to 5FU/Lv improved outcomes. This study together with two other studies resulted in oxaliplatin-containing regimens becoming the reference treatment in colon cancer stage III and in stage II with high risk features of recurrence. In a subgroup analysis of 315 patients aged 70–75 years old, no significant gain was seen in 6-year OS, 5-year DFS or 5-year TTR [68]. TTR was, however, similar in the 70–75 year age group and the younger age group, indicating that oxaliplatin added to 5FU prevents recurrences but does not translate into improved DFS or OS (Table 2).

In the NSABP C-07 trial [69] comparing bolus FU/FA ± oxaliplatin, a gain in DFS was seen for patients up to 70 years but not above. OS was worse in the elderly group. No difference in DFS was seen in patients aged older or younger than 65 years in the NO16968 trial (XELOXA-trial) [70].

Two of the three pivotal trials reported that the benefit of oxaliplatin was not seen in patients above age 70 years. A Cancer Care Ontario [71] systematic review of the evidence for benefit from adding oxaliplatin in patients older than 70 years of age found no benefit in any of the three trials, only greater levels of toxicity.

In a pooled analysis of four trials—NSABP C-08, XELOXA, X-ACT, and AVANT (bevacizumab was not considered)—DFS and OS were significantly improved in the patients who had been administered oxaliplatin irrespective of age, but less so in those above age 70 than in those below [72]. Toxicity was increased in elderly patients.

Similarly, a pooled analysis of seven trials comparing a newer adjuvant treatment, either a combination of drugs, usually oxaliplatin or an oral fluoropyrimidine with a standard 5FU/Lv regimen, patients aged 70 or above seemed to benefit less from adding oxaliplatin as regards OS and DFS [15]. For TTR, the difference between age groups was smaller, but younger individuals benefited to a greater extent.

In a third pooled analysis of the adjuvant trials performed during the past three decades and included in the ACCENT database [38], patients over age 65 years (not 70 as in the other analyses) benefitted from the addition of oxaliplatin in the trials including patients between 2004–2009 (NASABP C-08, PETACC-8, N0147, AVANT) but not in the trials (MOSAIC, NSABP C-07) which included patients between 1998–2003. In the PETACC-8 trial, randomizing colon cancer stage III patients to FOLFOX ± cetuximab [73], 10% of the patients above age 70 years did worse if they received cetuximab. Cetuximab did not have any favorable effect, only increased toxicity.

In a nomogram based upon the ACCENT database, age was an important factor for predicting OS, but was not included in the TTR nomogram [74], indirectly implying that the reduction in the risk of recurrence is not influenced by age whereas the ability to benefit from treatment is.
cancers-12-02289-t002_Table 2Table 2Retrospective analyses of the importance of age as regards the benefit of adjuvant chemotherapy in randomized trials and in pooled analyses of trials with patients treated for colon cancer.Reference StudyNumber of Elderly Patients/TotalTreatmentsAge Cut Off5-Year TTR BelowCut-Off5-Year TTR AboveCut-Off5-Year DFS BelowCut-Off5-Year DFS Above Cut-Off5-Year OS BelowCut-Off5-Year OS AboveCut-OffCommentsSargent et al. [12,13]7 trials506/3351FU/Lv vs surgery70
Similar

Statistically similar
Similar gains in elderlyGill et al. [12,13]7 trials 1864/3302FU/Lv vs surgery60

69% vs 56%*p* < 0.00163% vs 55%*p* = 0.000174% vs 67%*p* = 0,00269% vs 62%*p* = 0.0005Similar gains in elderlyQUASAR [63] 663/2291FU/Lv vs surgery (colon and rectum)70HR 0.78 (0.64–0.95)HR 1.13(0.74–1.75)

HR about 0.75HR 1.02No benefit above 70 years. TTR similar below 50, 50–59 and 60–69 yearsTaal et al. [64]279/324/1029FU/lev vs surgery60, 60–65 65–75

51% and 52% vs 61% and 58%48% vs 54%
64% and 60% vs 72% and 66%58% vs 66%No significant differences as regards ageTwelwes et al. [65]397/1987FU/Lv vs capecitabine70

54% vs 60%HR 0.8756% vs 58%HR 0.9763% vs 67%HR 0.9668% vs 73%HR 0.86No significant differences as regards ageShimada et al. [66]321/1092UFT/Lv vs 5-FU/l-Lv70

HR 1.02(0.80–1.30)HR 1.05(0.63–1.74)

Overall HR 1.02(0.84–1.23) No significant differences as regards ageYoshida et al. [67]536/1518UFT/Lv vs S170

71% vs 73% HR 0.93(0.73–1.19)68% vs 76%HR 0.73(0.56–1.02)

Overall HR 0.85(0.70–1.03) No significant difference as regards age.Tournigand et al. [39,68]315/2246FU/Lv vs FOLFOX470HR 0.74(0.62–0.88)70% vs 79% HR 0.72(0.47–1.11)HR 0.78(0.66–0.92)66% vs 69%HR 0.93(0.64–1.35)HR 0.80(0.66-0.97)76% vs 76%HR 1.10(0.73–1.65)No benefit in DFS/OS by adding oxaliplatin in atients 70+ yearsYothers et al. [69]396/2409FU/Lv vs FLOX70

65% vs 71%62% vs 63%HR 1.2(0.9–1.5)79% vs 82%76% vs 72%HR 1.3(1.0–1.7)No benefit from adding oxaliplatin in patients 70+ yearsHaller et al. [70]22% (≥70)/18865-FULV vs CAPOX65

HR overall 0.8 (0.7–0.9)HR similar as in younger pts, however, overlapping 1.0

XELOX better if < 65, ns. if ≥ 65Multivariate HR DFS 1.1 (1.0–1.2)/10 yr, OS 1.2 (1.1–1.3)/10 yrHaller et al. [72]4 trials 904/4819FU/Lv vs CAPOX/FOLFOX70

About 60% vs 72% HR 0.68(0.61–0.76)About 58% vs 68%HR 0.77(0.62–0.95)About 78% vs 85%HR 0.62(0.54–0.72)About 75% vs 78%HR 0.78(0.61–0.99)Assessed from K–M plots, 1921 pts FU/Lv, 2898 patients + oxaliplatinMcCleary et al. [15]7 trials1119/6539iv. 5-FU vs combination with oxaliplatin or oral capecitabine70HR 0.77 (0.69–0.85)HR 0.86 (0.69–1.06)*p*-interact = 0.32HR 0.78(0.71–0.86)HR 0.94(0.78–1.13)*p*-interact= 0.09 HR 0.83(0.74–0.92)HR 1.04(0.85–1.27)*p*-interact= 0.05Reduced benefit in elderly, no signs of interaction. Salem et al. [38]6 trials1754/6501FU/Lv vs FOLFOX/FLOX, old (1998–2003) vs new era trials (2004–2009)6572%70%70–71%64%78–80%99%Patients 65+ years did better when receiving oxaliplatin in new vs old era Taieb et al. [73]149/2559FOLFOX4 ± cetuximab70


HR 1.97(0.99–3.93)

Administering cetuximab to elderly worsened DFS, no gain in younger patientsAbbreviations: TTR = time to recurrence; DFS = disease-free survival; OS = overall survival; HR = hazard ratio; FU/Lv = 5-FU/leucovorin; ns = not significant; FOLFOX = 5-fluorouracil/leucovorin/oxaliplatin; FLOX = bolus 5-fluororuracil/leucovorin/oxaliplatin; CAPOX = capecitabine, oxaliplatin, i.v.= intravenous.


#### 3.1.3. Three vs. Six Months of Treatment with a Fluoropyrimidine and Oxaliplatin

A series of non-inferiority trials, summarized under the IDEA umbrella [75], have compared three and six months of treatment in 12,834 eligible patients with stage III colon cancer. For many patient groups, three months, at least with CAPOX and possibly also with FOLFOX was non-inferior to six months of treatment. Age was not addressed specifically in the IDEA publication, nor in the largest of the trials, SCOT [76]. In the TOSCA trial [77], no differences were seen between patients below or above 70 years of age (HR about 1.15 for both).

### 3.2. Rectal Cancer

#### 3.2.1. Fluoropyrimidines Alone

A Cochrane investigation of data from 20 older heterogeneous trials investigating adjuvant treatment in rectal cancer showed gains in either DFS (HR 0.75; 95% CI; 0.68–0.83) or OS (HR 0.83; 95%CI; 0.76–0.91) [78] (Table 3).

Since the quality of the staging, surgery, pathology, and preoperative therapy (RT/CRT rarely administered) represented older practices, these results have limited relevance today. Differences according to age were not described. Another Cochrane review showed similar gains in colon cancer stage II [86]. In the QUASAR trial [63], where pre- or post-operative RT was allowed, the recurrence risk was reduced in both rectal cancer (HR 0.68; 95% CI; 0.48–0.96) and colon cancer (HR 0.83; 95%CI; 0.65–1.07). A gain in DFS was seen independent of age (above/below 65 years, upper age limit usually 75) in a meta-analysis of 5 Japanese adjuvant trials testing oral uracil-tegafur (UFT) without RT [79].

Modern studies, where patients were pre-operatively treated with RT/CRT have greater clinical relevance today, and no firmly established gains have been seen in the individual studies or in meta-analyses summarizing the evidence [87,88]. No analyses according to age have been performed.

Thus, in the rectal cancer trials where patients are treated according to present routines and following guidelines (including administering neo-adjuvant therapy to risk groups [89] and surgery according to TME principles), no gains were seen from adjuvant chemotherapy (oxaliplatin added in one trial [90]). However, the studies were small and some prematurely interrupted because of poor patient inclusion. As in clinical practice, the compliance to the adjuvant treatment was generally poor (about 50%) [91,92]. Compliance was better in trials that randomized patients after surgery (about 70%).

Taken together, gains from adjuvant chemotherapy in rectal cancer are not evidence-based, although there is probably some gain not fundamentally different from that observed in colon cancer patients, considering that treatment initiation after diagnosis usually takes much longer in rectal than in colon cancer patients. It is possibly (one cannot use any stronger word considering the lack of proof from clinical studies) similar in elderly patients.

#### 3.2.2. Treatment with a Fluoropyrimidine and Oxaliplatin

Apart from the small negative Chronicle study [90], an oxaliplatin combination has not been tested as adjuvant therapy in rectal cancer without also adding oxaliplatin preoperatively. Several studies have incorporated oxaliplatin with a fluoropyrimidine concomitant with radiotherapy [80,85,93]. Although some gains in DFS were reported in two studies [80,85], the precise role of oxaliplatin cannot be evaluated due to the differences in 5-FU schedule and the addition of oxaliplatin both pre-operatively with radiation and post-operatively. Several trials have shown that the addition of oxaliplatin to chemo-radiotherapy is of no value [94,95,96,97,98].

Tumours with a pCR after pre-treatment have few recurrences and less need of adjuvant therapy [99,100,101]. In some studies, OS is apparently better in patients with pCR after chemo-radiotherapy who received adjuvant chemotherapy [102,103,104]; this is probably due to selection bias. A minor overall gain from adding oxaliplatin was restricted to patients whose tumours had an intermediate response to chemo-radiotherapy in the German CAO/ARO/AIO-04 trial [80]. Patients below age 61 years had the most pronounced gain, while patients between 61–70 had no significant gain and those above age 70 had no gain.

### 3.3. Information from Population Studies—Need for Randomized Studies in the Elderly

In population studies, better outcomes, particularly OS, have often been reported in adjuvant-treated patients [105,106]. However, the selection of fitter patients for treatment prevents proper conclusions, even after appropriate matching [107]. Population studies cannot confirm or refute the value of adjuvant therapy in CRC. If anything, they tend to overestimate its effects. Typically, in one meta-analysis of rectal cancer patients, the gains were restricted to non-randomized studies [108].

In four US Outcomes Databases, benefit from a fluoropyrimidine alone in stage III was suggested in patients above age 75 years, but the gains appeared to be overestimated (propensity scored HR for mortality 0.42–0.76) [109]. In a SEER/Medicare [110] database study, no benefit could be detected in patients older than 65 years with stage II malignancies, even if high risk features were present. In a similar analysis of four US effectiveness population cohorts, the authors suggest that the use of oxaliplatin added to 5FU improves OS at three years in patients between 65 and 75 years in the community, contrasting with what other trials have shown [111,112]. In a Japanese study including 4,598 patients from several hospitals, 21% of which were older than 75 years [113], elderly patients received adjuvant therapy much more seldom than younger ones, but treated patients fared better, the results resembling those reported in studies in the Western World.

Randomized trials are needed to identify the comparably limited gains seen from administering adjuvant chemotherapy and, in particular, the additional effects provided by the addition of oxaliplatin. The randomized PRODIGE 34-FFCD 1402-ADAGE study, (ClinicalTrials.gov Identifier: NCT02355379) [114], recruits patients as of 24 April 2020. A Chinese randomized phase III trial (ClinicalTrials.gov Identifier: NCT02316535) in elderly patients was activated in 2015, with no information since 2016. It aimed at comparing capecitabine in a standard dose of 1250/m^2^ × 2 × 14 with a low dose, 1000 mg/m^2^ × 2 × 14 in patients aged 70–90 years of age. In most Western countries, the usual dose to elderly patients is 1000 mg/m^2^ and not 1250 mg/m^2^, and in elderly frail patients, even lower doses are initially chosen.

## 4. Is the Timing of Adjuvant Treatment Important as Regards Efficacy?

The gains in colon cancer stage III from modulated 5-fluorouracil (improved DFS by 35%) were seen in trials where the delay to treatment initiation was short (generally less than five weeks). In two parallel Scandinavian adjuvant chemotherapy trials [115], one-third of patients started treatment later than eight weeks, and no statistically significant gain was seen in colon cancer stage III patients. In the patients treated at Norwegian centres, where therapy was to start within 3–4 weeks, and at the latest within six weeks, a significant gain was seen [116]. In the trials proving that the addition of oxaliplatin improved DFS (HRs about 0.8), the allowed delay was seven weeks in [69,117] and eight weeks in [70].

In routine clinical practice, adjuvant treatment is usually started later, and the relevance of this has been the subject of many retrospective studies. Associations with OS have been found in several of them [118,119,120,121,122] and, although plausible, it has not been established if a reasonable delay is deleterious. In the XELOXA-trial [70], every 10-day delay from surgery to randomization significantly decreased both DFS (HR 1.09 (1.01–1.17)) and OS (HR 1.10 (1.00–1.21). It has not been reported if adjuvant treatment is initiated later in elderly patients, although this is possible since the post-operative course may be longer.

This “rapid” initiation has not been possible in any rectal cancer trial. This allows the subclinical deposits, to be eradicated by the adjuvant treatment, time to proliferate a few months longer during neo-adjuvant treatment. The deposits may then contain so many tumour cells that they cannot be completely eradicated. The response to chemotherapy varies and in highly sensitive and resistant tumours, time may be irrelevant, but for those with intermediate sensitivity, a few weeks delay may be detrimental.

## 5. Neo-Adjuvant Rather than Adjuvant Therapy?

Adequate chemotherapy pre-operatively rather than post-operatively should result in fewer systemic recurrences if time matters. Three cycles of neo-adjuvant mFOLFOX and nine adjuvant cycles resulted in fewer recurrences at two years of primary colon cancer compared with 12 post-operative cycles in the FOxTROT trial (14% vs. 17.5%; HR 0.74; 0.54–1.00) [123]. In a subgroup analysis according to age, a tendency towards a reduced recurrence rate was also seen in elderly patients (70+ years, HR 0.61, 0.35–1.07; 60–69 years, HR 0.68, 0.42–1.09; 50–59 years, HR 0.77, 0.40–1.48).

In rectal cancer, short-course RT followed by three cycles of FOLFOX was not better than chemo-radiotherapy; adjuvant chemotherapy could be given to both groups [83,84]. However, three cycles of FOLFOX may be too few to reduce the distant metastasis rate. In the RAPIDO trial [81], patients with high-risk rectal cancer were randomized to chemo-radiotherapy, surgery, and optional adjuvant chemotherapy or to short-course RT followed by six courses of CAPOX and surgery. A reduction in the primary endpoint, disease-related treatment failure (HR 0.75;0.60-0.96), was seen regardless of patient age [82]. This approach (scRT and CAPOX followed by surgery) can be considered a new reference treatment in high-risk rectal cancer. Neo-adjuvant chemotherapy has been added as one treatment option in recent guidelines [124] and is used increasingly, especially to facilitate organ preservation after both conventional CRT and short-course RT [125,126,127,128,129].

## 6. Toxicity to Adjuvant Chemotherapy in Elderly Patients: Importance of Geriatric Assessments

Although it is important to properly evaluate every cancer patient prior to treatment decisions in order to individualize therapy, this is particularly important in older adults since functional reserves are decreased and co-morbidities frequent. The variability in the abilities to tolerate and accept chemotherapy is pronounced and increases with age. A geriatric assessment (GA) is considered mandatory in order to evaluate the biological age and life expectancy of the individual [130]. The regularly used performance status evaluations are insufficient [131,132].

In a report of 120 patients treated at one hospital during a 10-year period, no increased toxicity could be noted in patients between 65–75 years when treated with 5-FU-based adjuvant therapy [6]. In another hospital-based series of 529 patients treated with a fluoropyrimidine alone or with oxaliplatin, increased toxicity was seen despite significant dose reductions in patients above 70 years, but tumour outcomes were not negatively affected. In a Dutch study including 216 patients above 75 years treated with 5FU, increased toxicity was seen, but also better survival than in non-treated patients [133].

The increased toxicity to oxaliplatin-containing chemotherapy seen in the elderly usually fit patients included in the trials was also observed in a population-based study of Dutch patients above 70 years of age [134]. Grade III–IV toxicity had a pronounced influence on cumulative dosages received and completion of all planned cycles, this explaining why the elderly do not appear to benefit from the addition of oxaliplatin. The administration of CAPOX or capecitabine was associated with better RFS and OS but the completion of treatment was more important than the choice of regimen [134]. The authors recommended capecitabine alone rather than CAPOX. In another study of older patients, even lower grade toxicities (less than grade III–IV) influenced compliance to treatment [135], further emphasizing that adding oxaliplatin is of questionable value in elderly patients.

Attempts have been made to predict chemotherapy toxicity in older adults [136] and it is possible to use stratification schedules in clinical practice to individualize therapy.

The International Society for Geriatric Oncology (SIOG) recommends patients with cancer above an age of about 70 to have a GA [2,131]. A comprehensive GA (CGA), i.e., a multi-faceted evaluation of general well-being of elderly, is preferably recommended [2,137]. It identifies health issues that may otherwise be routinely overlooked (see [131] for a review). The involvement of a geriatrician is claimed to grant the opportunities to identify co-morbidities for optimization and to address psychological needs. A GA/CGA may help in the decision of a particular treatment [138], but must be followed by appropriate interventions to help the patient.

## 7. Conclusions

Even in the absence of randomized trials recruiting elderly patients in order to demonstrate that adjuvant chemotherapy reduces recurrence rates after colon cancer surgery to such an extent that DFS and possibly also OS are improved, multiple retrospective analyses of the rather limited number of elderly patients (typically 15% of the total material to hand) indicate that fit elderly patients have the same gain from six months of fluoropyrimidine treatment as younger adults. SIOG considers adjuvant 5FU-based chemotherapy in elderly patients well established in stage III [2] disease, similar to the situation in younger patients. Similarly, treatment may also be motivated in stage II if risk factors for recurrence are present. The gains are not pronounced, HR about 0.7, and, for many elderly patients with a comparably limited risk of recurrence, insufficient for routine therapy. Like in young patients, MSI indicates less risk of recurrence, at least in stage II, and adjuvant therapy is not motivated.

It is questionable if elderly patients (above approximately 70 years) have any gain from the addition of oxaliplatin to the adjuvant regimen since none of the three trials directly indicated that they benefit sufficiently. In an article from SIOG [2], it is stated that “the gains from the addition of oxaliplatin for up to six months are modest and most of the benefit is still conferred by the fluoropyrimidine”. It is up to an individual evaluation of the patient by the physician, taking into consideration the risk of recurrence and the increased toxicity from combined treatment in relation to the biologic age of the patient. The many pooled retrospective analyses, including trials not directly randomizing patients between a fluoropyrimidine ± oxaliplatin, have indicated that elderly patients might benefit, although the magnitude of benefit is at best limited (HR about 0.8). Since toxicity to oxaliplatin with a fluoropyrimidine is a major concern in elderly patients, three months of treatment is an attractive option, even in high-risk groups for recurrence such as those with pT4 and pN2 tumours. This has simplified the decision in individual patients, but for most elderly patients, a fluoropyrimidine alone for six months remains the recommended adjuvant therapy.

In rectal cancer, the knowledge base is weak, and strong evidence of sufficient gains from adjuvant chemotherapy is not even present in young adults. One reason may be a longer delay from diagnosis to treatment initiation, caused by a long pre-treatment period with a delay to surgery. This is most probably also true in elderly patients, but no data are available.

Since neo-adjuvant chemotherapy is generally more tolerable than post-operative chemotherapy and, according to recent trials in both colon and rectal cancer, also more effective in reducing recurrences (improved DFS or a comparable endpoint, DrTF), this can be recommended for elderly patients despite the absence of trial data.

## Figures and Tables

**Table 1 cancers-12-02289-t001:** Censored recurrence risks (RR) at 3 and 5 years in the entire Swedish colon cancer population from 2007–2016

Stage		No Adjuvant Treatment			Adjuvant Treatment		
	Age	*N*	3y RR	5y RR	*N*	3y RR	5y RR
Stage II	*<50*	204	3%	3%	131 (39%)	6%	9%
	*50–59*	502	6%	8%	217 (30%)	13%	14%
	*60–69*	1620	8%	10%	475 (23%)	13%	15%
	*70–79*	3139	8%	10%	432 (12%)	14%	17%
	*80–89*	2632	7%	9%	49 (2%)	16%	16%
	*>90*	281	8%	9%	0		
Stage III	*<50*	7	29%	29%	394 (98%)	25%	28%
	*50–59*	43	24%	24%	662 (94%)	19%	22%
	*60–69*	178	35%	40%	1762 (91%)	23%	27%
	*70–79*	816	27%	31%	1876 (70%)	26%	30%
	*80–89*	1562	27%	31%	227 (13%)	33%	39%
	>90	177	23%	28%	0		

N = number of patients; y = year; RR = recurrence risk.

**Table 3 cancers-12-02289-t003:** Retrospective analyses of the importance of age for the benefit of adjuvant/neo-adjuvant chemotherapy in randomized trials in patients with rectal cancer.

Study Randomization Groups	Number of Elderly Patients	Treatment	Age Cut Off	5-Year DFS Below	5-Year DFS Above	5-Year OS Below	5-Year OS Above	Comments
QUASAR rectal [63]	totally 948 patients	Control vs. FULv	70					No information about age for rectal cancer patients
Sakamoto et al. [79]	541/2091 in 5 trials	Control vs. UFT	65	HR 0.73	HR 0.72	HR 0.79–0.82	HR 0.88	Overall benefit DFS/OS, no difference according to age
CAO/ARO/AIO-04 trial, Rödel et al. [80]	471, 60–69 294, 70+, years totally 1236 pts	CRT and adjuvant treatment ± oxaliplatin	60, 70	−61: HR 0.61 (0.43–0.86), 61–70 HR 0.87 (0.61–1.24)	70+: HR 1.06 (0.71–1.58)			No benefit in elderly patients
RAPIDO [81,82]	362/912	CRT + optional adjuvant CAPOX vs. scRT + neo-adjuvant CAPOXx6	65	HR 0.75	HR 0.75			Reduced DrTF in favour of neo-adjuvant chemo in both age groups
Polish II trial [83,84]	not reported/515	CRT vs. scRT + neo-adjuvant FOLFOX4x3						No data on age
Adore [85]	87/321	CRT and adjuvant treatment ± oxaliplatin	65	HR 0.64 (0.42–0.97)	HR 0.60 (0.21–1.67)			Similar HRs for elderly patients

Abbreviations: Control = observation after surgery; FLv = 5-fluorouracil/leucovorin; cap=capecitabine; FOLFOX = 5-fluorouracil/leucovorin/oxaliplatin; CAPOX = capecitabine, oxaliplatin; scRT = short-course radiotherapy, 5 × 5 Gy in one week; CRT = conventional chemoradiotherapy to approximately 50 Gy with a fluoropyrimidine; ACT = adjuvant chemotherapy; TTR = time to recurrence; DFS = disease-free survival; DrTF = disease-related treatment failure (used in the RAPIDO trial); OS = overall survival; HR = hazard ratio.

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
