# Peer review of "Adjuvant Chemotherapy in Elderly Colorectal Cancer Patients"

_cancers, 2020, doi:10.3390/cancers12082289_

Round 1

Reviewer 1 Report

The authors have done a nice job reviewing the literature on adjuvant therapy for elderly patients with colorectal cancer. This is a very interesting topic, largely due to lack of dedicated prospective studies for this patient population. The authors point out all the challenges, including what age defines “elderly”, the accurate endpoint we should measure success with, outcome differences for the elderly in the past and present, and the differences in benefit for colon and rectal cancers. A few comments and questions for the authors:

-Do you attribute the improved current outcome (TTR and DFS) for elderly patients treated in the current era due to improvement in chemotherapy delivery? There is comment about “improved possibilities of treating elderly patients with an oxaliplatin combination” (line 106). Does this mean that the addition of oxaliplatin has been the reason for the improved outcome? Or, do you mean that we are able to give chemo safer and in a more tolerable manner? eg altering 5FU bolus, improved anti-emetic regimens.

-I think the difficulties with rectal cancer, as the authors have pointed out is that in rectal cancer trials, although the patients may be randomized to treatment arms to receive adjuvant chemotherapy- the number of patients who are able to receive it are low. Thus, it is difficult to ascertain the true benefit but it does not mean systemic chemo is not effective. I think with the neoadjuvant studies coming out- it may be important to see whether triplet chemo (FOLFIRINOX) vs doublet chemo (FOLFOX) should be offered to elderly patients.

-Would like to point out that for stage II MSI-high colon cancer patients do not benefit from 5FU adjuvant therapy, there is still data that stage III MSI-high colon cancer patients do benefit from FOLFOX.

-The authors have presented data on the Swedish population (Table 1), and they have summarized data from stage II patients who received adjuvant therapy. They state that recurrence risk is worse for the elderly patients who received adjuvant therapy vs those who did not- because they had higher-risk features. Do we know that that all the stage II patients over the age of 50 had high risk features (what were they) or is that an assumption? Any way to know BRAF, MSI status? Do we know if adjuvant therapy actually decreased their recurrence risk? Although the N is different in the 2 populations, stage III patients did not seem to have improvement in 5y RR with adjuvant therapy- or was this patient selection?

-In the conclusion, would like for authors to incorporate a few other thoughts about elderly patients and adjuvant therapy. How are they incorporating MSI and BRAF mutation status? Do they recommend adjuvant 5FU therapy for high-risk Stage II eg T4? Are they recommending 6 months of 5FU therapy or 3 months of XELOX for stage III?

-Some grammer review eg line 6, line 81, line 116

Author Response

Reviewer 1

In response to reviewer 1 regarding the improved possibilities to treat patients with chemotherapy and especially with oxaliplatin (paragraph 1): Most of the improvements in the current era are due to other improvements, e.g. better surgery, preoperative evaluations, i.e. not to the use of chemotherapy. The quotation by the reviewer relates to the finding in the most recent ACCENT data base summary, and not to the two previous summaries from that group. The general health and quality of care for several other diseases have also improved impacting survival. The ability to safely deliver oxaliplatin impacts both survival and recurrence rates, more patients can be considered candidates for oxaliplatin. So, the answer to the last question is both.

We agree with the reviewer that in rectal cancer (paragraph 2), we don´t know whether adjuvant chemotherapy works or does not work. In the most fit elderly, it is important to provide even more effective adjuvant therapy.

Thank you for the comment (paragraph 3). The literature is inconclusive, no one knows for sure, but we agree.

Regarding the comments in paragraph 4, there is no data in the national SCRCR regarding MSI and BRAF status for these patients. In a previous publication from the registry regarding the population about 16% in stage II and 20% percent of stage III patients receiving chemotherapy had no NCCN-risk factor (pT4, pN2, emergency surgery, high-grade malignancy, vascular or perineural invasion, and <12 sampled lymph nodes). The apparent “lack of effect” from chemotherapy is due to patient selection (more patients having high risk features). The effect of chemotherapy brings their recurrence data in line with the patients with low risk features. It is in our view not possible to draw conclusions about efficacy from adjuvant chemotherapy from populations where the treatment has been given selectively.

Concerning the many comments and questions in paragraph 5, our opinion is that to be considered for chemotherapy, the elderly patient must be fit and of reasonable biological age. We devoted part of the manuscript to the question if recurrence risks differ between young and elderly and concluded that they don´t. Thus, MSI-status should be handled similarly, i.e. in stage II indicating low risk of recurrence and no solid evidence of enough need or benefit. The knowledge about how to handle BRAF-mutated tumours is limited; much overlap is also seen between MSI-H and BRAF-mt. In stage II with more than one risk factor or stage III without risk factors it is probably reasonable to offer 5-FU/Capecitabine. In stage III with risk, 3 months of oxaliplatin containing adjuvant treatment may be offered but considering the lack of firm evidence in elderly patients and the potential toxicity to oxaliplatin, 5-FU/Capecitabine for 6months is a similarly good alternative in the elderly.

It is difficult to give any firmer recommendations than we already did in the original text emphasizing the aspects brought up by this reviewer. We have, however, added a few sentences and slightly modified the text (rows 406-421). The two added sentences read as follows: “Similarly, treatment may also be motivated in stage II if risk factors for recurrence are present.” and “Like in young patients, MSI indicates less risk of recurrence, at least in stage II, and adjuvant therapy is not motivated.” We consider 6 months of a fluoropyrimidine as more often relevant for the group of elderly patients than 3 months of an oxaliplatin combination, but this must be discussed individually (as most in medicine).

Changes to grammar as suggested by reviewer 1

Line 6: Is an email address. We are not sure which line the reviewer meant.

Line 81: changed suggests to suggest

“recent evidence suggest that recurrence risks, clinically dominated by distant metastases, have decreased”

Line 116: Has been rewritten and changed to: The risk of (systemic) recurrence is about 20% in multicentre series [43] and in nationwide populations [42] when patients are adequately staged and treated, providing the potential gain from adjuvant treatment.

Reviewer 2 Report

This is a well written and comprehensive report dealing with the adjuvant treatment of colon cancer in the elderly population.

Authors should recognize that the contribution of geriatric assessment is directed also to determine the life expectancy irrespective of the presence of colon cancer.

In addition,Authors suggest that three months of capox should be considered a worthwhile option for high-risk stage III pts (ie T4 and/or N2)

Albeit not supported by data generated by clinical trials,three months of capox followed by three months of fluoropyrimidine alone could be a reasonable option as well.

Author Response

Reviewer 2

In response to reviewer 2 we have added the underlined wording on line 346/347:

“A geriatric assessment (GA) is considered mandatory in order to evaluate the biological age and life expectancy of the individual”

3 months of CAPOX followed by 3 months of a fluoropyrimidine could, as the reviewer note, be an attractive option in elderly patients also in stage III with high risk features, however, as he/she also notes, there is a lack of data supporting it. Six months might be needed to reduce recurrence risks as much as possible, but the collected evidence so far in elderly is that the benefit of adding (up to 6 months) oxaliplatin is limited. We have chosen not to modify the manuscript since the present uncertainties were already incorporated. Prompted by reviewer 1, we believe that for most elderly patients fit enough for treatment, 6 months of a fluoropyrimidine is likely a more attractive alternative than e.g. 3 months of an oxaliplatin combination. We clarified this slightly more on line 422 by adding the words “for 6 months”.